systems biology/computational biology/
biomathematics

cell cycle, circadian clock, coupled oscillators,
synchronization ratios, modelling and simulation,
control parameters

**Author for correspondence:**
S. Almeida
e-mail: sofia.jf.almeida@gmail.com

# Control of synchronization ratios in clock/cell cycle coupling by growth factors and glucocorticoids

## S. Almeida[1], M. Chaves[2] and F. Delaunay[3]

[1]Université Côte d'Azur, iBV, Inria, Biocore team, Sophia Antipolis, France
[2]Université Côte d'Azur, Inria, INRA, CNRS, Sorbonne Université, Biocore team, Sophia Antipolis, France
[3]Université Côte d'Azur, CNRS, INSERM, iBV, Nice, France

SA, 0000-0002-7547-8784; MC, 0000-0003-0647-9254;
FD, 0000-0003-4927-1701

The cell cycle and the circadian clock are essential cyclic cellular processes often synchronous in healthy cells. In this work, we use previously developed mathematical models of the mammalian cell cycle and circadian cellular clock in order to investigate their dynamical interactions. Firstly, we study unidirectional cell cycle → clock coupling by proposing a mechanism of mitosis promoting factor (MPF)-controlled REV-ERBα degradation. Secondly, we analyse a bidirectional coupling configuration, where we add the CLOCK:BMAL1-mediated MPF repression via the WEE1 kinase to the first system. Our simulations reproduce ratios of clock to cell cycle period in agreement with experimental observations and give predictions of the system's synchronization state response to a variety of control parameters. Specifically, growth factors accelerate the coupled oscillators and dexamethasone (Dex) drives the system from a 1 : 1 to a 3 : 2 synchronization state. Furthermore, simulations of a Dex pulse reveal that certain time regions of pulse application drive the system from 1 : 1 to 3 : 2 synchronization while others have no effect, revealing the existence of a responsive and an irresponsive system's phase, a result we contextualize with observations on the segregation of Dex-treated cells into two populations.

# 1. Introduction

The cell division cycle and the circadian clock are two fundamental cyclic processes of cellular control that tend to be synchronous in a variety of healthy cell types. In the process of cell growth and division, the cell undergoes a sequence of observable changes culminating in mitosis, a process that is then

restarted by daughter cells. The circadian clock is a biological oscillator conserved across species that results in cell-autonomous 24 h rhythms (circadian rhythms). In mammals, peripheral cellular clocks are entrained through internal synchronizers by a central pacemaker localized in the suprachiasmatic nuclei of the hypothalamus; these peripheral clocks generate circadian patterns of gene activation and protein expression at the cellular level [1,2].

Both clock and cell cycle processes are essential for cellular health in mammals and when unregulated can result in disease at the organism level. In particular, cancer is characterized by unregulated growth of cells that have accumulated driver mutations and genetic or environmental clock disruption has been shown to promote malignant growth [3]. Owing to the tight interconnection between the two oscillators, deregulation in one of them potentially deregulates the other as well, as evidenced by increased risk of circadian clock disturbances in cancer patients [4]. Because both systems result in rhythmic behaviour they can be interpreted and modelled as oscillators, subjected to some form of coupling.

Several molecular interactions revealing direct action of the clock on the cell cycle have been discovered. Firstly, the CLOCK : BMAL1 protein complex, essential for the circadian clock, induces expression of the *wee1* gene [5]. The kinase WEE1 phosphorylates and inactivates the CDK1 and CDK2 kinases, thus inhibiting the essential cell cycle complex cyclin B-CDK1, or mitosis promoting factor (MPF). Furthermore, the clock components REV-ERB-$\alpha/\beta$ (nuclear receptors) and ROR-$\alpha/\gamma$ (retinoic acid-related orphan receptors) regulate the cell cycle inhibitor p21 [6]. Finally, there is also evidence for clock repression of c-Myc, a promoter of cell cycle progression by cyclin E induction [7], that is deregulated in mice deficient in the gene encoding for the core clock protein PER2 [8].

The observation of circadian rhythms of cell division in a variety of organisms [4] first led to an hypothesis of 'gating' of the cell cycle by the clock mechanism [3], which considered clock control of the cell cycle to only allow mitosis to occur at certain time windows. An example of a model of cell cycle gating by the clock is provided by Zámborszky *et al.* where critical size control of the mammalian cell cycle was found to be triggered by the clock [9]. By contrast, Gérard and Goldbeter simulate entrainment of the cell cycle by the clock, while also suggesting a possible form of gating by the clock at the entry of G1 phase through a mechanism of oscillating growth factor (GF) [10].

Contrary to previous observations showing mostly an unidirectional action of the clock on the cell cycle, Feillet *et al.* and Bieler *et al.* have demonstrated phase-locking between clock and cell cycle with strong evidence for bi-directional coupling [11,12]. Phase locking is characterized by convergence of the combined phase of oscillation $\phi(t) = (\phi_1(t), \phi_2(t))$ to a closed curve—an attractor. This differs from the gating model, as phase-locked oscillators are synchronized through the entire cycle—knowing the phase of one oscillator determines the phase of the other, in ideal noise-free systems. By contrast, in the gating hypothesis only the mitotic phase would have to align with specific clock phases.

The present work is motivated in large part by observations of Feillet *et al.* on phase-locking between the cell cycle and the circadian clock of mammalian cells [11]. The authors have observed that increasing the concentration of GFs (expressed as % of fetal bovine serum (FBS)) in the growth medium of NIH3T3 mouse fibroblasts results not only in an expected increased cell cycle frequency but also in an equal trend of increase in clock frequency [11], such that the two oscillators always remain synchronized in a 1 : 1 period ratio for different values of FBS.

Furthermore, Feillet *et al.* observed the phase-locking behaviour of cells under the application a of a pulse of dexamethasone (Dex), a synthetic glucocorticoid agonist known to synchronize clocks in populations of mammalian cells [11]. This application resulted in different clock to cell cycle period ratios depending on the concentration of GFs [11]. These synchronization ratios in Dex-treated cells were determined to be approximately 5 : 4 for 10% FBS and 3 : 2 for 20% FBS [11]. Additionally, cells grown in 20% FBS segregate into two groups, one with 3 : 2 synchronization and the other with cells remaining in 1 : 1 phase-locking (just as without Dex application). From these results as well as mathematical modelling, the authors conclude the existence of multiple attractors for clock and cell cycle phase-locked behaviour [11], i.e. that the Dex input may be shifting the oscillators from one limit-cycle to another.

Moreover, Feillet *et al.* have verified that for 1 : 1 phase-locking the cell cycle division occurs at a specific clock phase for all cells, while the synchronization dynamics of the second group of 20% FBS after Dex-treatment shows a trimodal frequency peak distribution of mitosis with circadian clock phase. A similar observation of trimodal peak distribution had previously been made by Nagoshi *et al.* under a similar protocol [13]. Furthermore, Bieler *et al.* have obtained similar results of clock/cell cycle 1 : 1 phase-locking with the increase of GFs [12]. While the three peak distribution of cell density by itself does not exclude the 'gating' hypothesis, observations of 1 : 1 phase-locking support the hypothesis of coupled oscillators and suggest that coupling from the cell cycle to the circadian clock

occurs in mammalian cells. Thus, there is probably bidirectional coupling between the oscillators. Our work aims at gaining insight on dynamical mechanisms that may be behind the observations of Feillet *et al.* [11], in particular, the different synchronization ratios, and explore uni- and bi-directional coupling.

Precise knowledge on how the cell cycle may influence the clock isn't available. Cell cycle → clock coupling has been studied by Traynard *et al.* in a modelling work with the incorporation of transcription regulation of clock genes during the mitosis stage [14]. The authors focus on transcription regulation, proposing a role of MPF in the transcription activation of clock genes at the mitosis stage by use of a periodic coupling parameter. Moreover, there is experimental evidence that the transcription factor c-Myc, known to favour cell cycle progression, has also a role in clock regulation [15,16].

In this paper, we explore the idea of phosphorylation of an essential clock component by MPF. Specifically, we focus on the MPF-mediated REV-ERBα phosphorylation and subsequent degradation, which is an experimental observation [17]. Notably, this is supported by the observation that cells arrested in G2 upon treatment with the microtubule depolymerization drug nocodazole show a significant decrease in REV-ERBα abundance. On the other hand, mechanisms denoting a clock influence on the cell cycle are a consensual observation. Of these, CLOCK : BMAL1 promoting the MPF repressor *wee1* [5] is a mechanism that involves the essential cell cycle and clock complexes (MPF and CLOCK : BMAL1) and is the one we study here.

We focus specifically on states of synchronization of the two oscillators with respect to their period, or period-locking (PL). Phase-locking (that implies period-locking) is also obtained, but the analysis will largely focus on the evolution of the ratio of clock to cell cycle period $r_T$ under different forms of coupling mechanisms, added inputs and parameter changes. Furthermore, the effect of Dex is a particularly relevant subject explored in this work as it relates to the experimental observations of Feillet *et al.* [11].

Thus, in §3, we analyse the cell cycle → clock unidirectional coupling, by modelling the MPF-induced degradation of REV-ERBα [17]. We are able to recover not only the 1 : 1 and the 3 : 2 period-lock ratios, but also the experimentally observed effect of Dex-treatment in inducing a change of synchronization ratio. Following this, we study, in §4, the bi-directionally clock ⇆ cell cycle coupled system, where the aforementioned interaction is combined with the known molecular interaction whereby CLOCK : BMAL1 indirectly represses MPF by promoting the *wee1* gene [5]. Finally, we analyse the effect of a Dex pulse (as opposed to a constant Dex input) in bidirectional coupling and find the time of pulse application $T_{\text{pulse}}$ to be a control parameter for the system's synchronization response.

# 2. Methodology

In this work, we study the coupling of the mammalian cell cycle reduced model developed by Almeida *et al.* [18] with the mammalian circadian clock model developed by Almeida *et al.* [19] and investigate unidirectional cell cycle → clock coupling as well as bidirectional coupling. Recently, Almeida *et al.* [20] have used the same models to study period control methods in unidirectional clock → cell cycle coupling via CLOCK : BMAL1 induction of *wee1*, an interaction we will also add to our system in §4.

The two variable cell cycle model includes the fundamental negative regulatory loop between the MPF, cyclin B-CDK1, with its repressor the APC : cdc20 complex, as well as the positive self-regulatory loops of MPF with itself representing its actions in activating its activator CDC25 and de-activating its repressor WEE1 [18]. This model produces relaxation oscillations whose period is controlled by the GF input. It is given by the equations

$$\frac{\mathrm{d}[\text{MPF}]}{\mathrm{d}t} = \text{GF} + V_c \frac{\overline{\text{MPF}_{\text{max}}} - [\text{MPF}]}{\overline{\text{MPF}_{\text{max}}} - [\text{MPF}] + k_c} \frac{[\text{MPF}]^2}{[\text{MPF}]^2 + k_m^2}$$
$$- V_w \frac{[\text{MPF}]}{[\text{MPF}] + k_w} \frac{k_n^2}{[\text{MPF}]^2 + k_n^2}$$
$$- \gamma_1 [\text{APC} : \text{cdc20}][\text{MPF}] \tag{2.1}$$

and

$$\frac{\mathrm{d}[\text{APC} : \text{cdc20}]}{\mathrm{d}t} = V_m [\text{MPF}] - V_k [\text{APC} : \text{cdc20}]. \tag{2.2}$$

**Table 1.** Parameters of the models for the two oscillators after scaling.

| $p$ | numerical value |
|---|---|
| $V_R$ | 34.4 h$^{-1}$ |
| $k_{Rr}$ | 80.1 |
| $V_B$ | 0.11 h$^{-1}$ |
| $V_{D2}$ | 14.7 h$^{-1}$ |
| $\gamma_{rev}$ | 0.187 h$^{-1}$ |
| $\gamma_{db}$ | 0.121 h$^{-1}$ |
| $\gamma_{bp}$ | 2.0 h |
| $\gamma_1$ | 0.162 h$^{-1}$ |
| $V_c$ | 2260 h$^{-1}$ |
| $k_c$ | 130 |
| $V_w$ | 7480 h$^{-1}$ |
| $k_w$ | 138 |
| $k_m$ | 99 |
| $k_n$ | 0.116 |
| $V_m$ | 0.168 h$^{-1}$ |
| $V_k$ | 0.107 h$^{-1}$ |
| $\overline{MPF}_{max}$ | 284 |

The clock model is based on transcriptional regulation and is able to recover the antiphasic relation in the oscillation of the CLOCK : BMAL1 and PER : CRY proteins [19]. We begin by dynamically reducing this model in order to obtain the smallest possible network that maintains this property. This is done in appendix A and results in a four variable model:

$$\frac{d[BMAL1]}{dt} = V_R \frac{k_{Rr}^2}{k_{Rr}^2 + [REV]^2} - \gamma_{bp}[BMAL1][PER : CRY] \tag{2.3}$$

$$\frac{d[DBP]}{dt} = V_B[BMAL1] - \gamma_{db}[DBP] \tag{2.4}$$

$$\frac{d[REV]}{dt} = V_{D2}[DBP] - \gamma_{rev}[REV] \tag{2.5}$$

and
$$\frac{d[PER : CRY]}{dt} = V_{D2}[DBP] - \gamma_{bp}[BMAL1][PER : CRY]. \tag{2.6}$$

Original parameters for both models were obtained by nonlinear optimization based on cost-minimization, with the use of a MATLAB function [18,19]. In order to have oscillation of both systems with periods of the same order of magnitude and consistent with values of experimental observations (namely $T_{clock} = 24$ h) we scale the original parameters of both models, by multiplying those referring to rates of change by an appropriate constant. As both systems were previously normalized to a certain concentration value, the solution of the coupled system is dimensionless. Table 1 shows the final parameter values (except the parameter GF that will be varied during this study). Additionally, for all simulations, we use the initial condition: BMAL1 = 1.2; DBP = 1.6; REV = 1.5; PER : CRY = 1.2; MPF = 2.0; APC : cdc20 = 1.0. This condition was tentatively chosen.

# 3. Unidirectional coupling via mitosis promoting factor-induced phosphorylation of REV-ERB$\alpha$

Following the discovery made by Feillet *et al.* and Bieler *et al.* we know that in NIH3T3 cells the oscillators show 1 : 1 phase-lock and that the influence of the cell cycle on the clock seems to be as relevant as the reverse [11,12]. Thus, we begin by studying the unidirectional cell cycle → clock coupling. For this, we consider the MPF phosphorylation of REV leading to its subsequent degradation [17], as shown in figure 1.

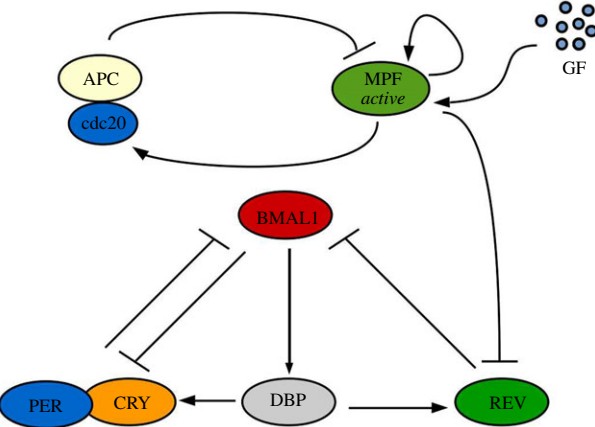

**Figure 1.** Schematic of the unidirectional cell cycle → clock coupling mechanism. MPF in its active form represses REV via phosphorylation leading to REV's subsequent degradation.

As such, we multiply the degradation term of REV by $c_m$ [MPF], where $c_m$ is a constant, representing the coupling strength. The equation for the REV rate of change becomes

$$\frac{d[\text{REV}]}{dt} = V_{D2}[\text{DBP}] - c_m \gamma_{\text{rev}}[\text{MPF}][\text{REV}]. \tag{3.1}$$

As MPF has an enzymatic activity, catalysing the reaction without being consumed in it, we consider its rate of change to be unaffected by this interaction.

## 3.1. The effect of growth factor

We start by verifying that entrainment of the clock by the cell cycle is possible for strong coupling ($c_m =$ 0.2), see figure 2. Both oscillators maintain overlapping periods and the ratio of clock to cell cycle period $r_T = T_{\text{clock}}/T_{\text{cell cycle}}$ is always 1.

Conversely, for weak/moderate coupling, we verify different values for the period-lock ratio $r_T$, where ratios between the two periods follow a specific pattern with increasing GF. Period-lock ratios remain constant at rational values for some GF intervals. This pattern has visual similarity to that of the *devil's staircase* fractal curve [21]. Figure 3 shows results for $c_m = 0.04$ (top) and $c_m = 0.08$ (bottom). We verify that different $c_m$ values induce different $r_T$ values and that non-integer rational ratios are present, including the experimentally observed 3:2 period-lock in both cases [11]. Increasing GF increases $r_T$, while increasing $c_m$ has the opposite effect, causing a shift of the point where the system is driven from 3:2 to 2:1 synchronization to a higher GF value.

A similar observation holds for $c_m$ as seen in figure 4, which shows the system's period-lock response to variation of this control parameter for fixed GF.

In figures 3 and 4, there are some points that fall outside the *devil's staircase* pattern, which is owing to complex behaviour. The electronic supplementary material, S1 provides a brief overview of our period measurement algorithm, which measures the number of relevant peaks. Moreover, following the results of figure 4, electronic supplementary material, figures S2 and S3 show oscillatory solutions for fixed GF = 40 with $c_m = 0.08$ and $c_m = 0.04$, respectively, illustrating 3:2 and 2:1 period-locked oscillations. The electronic supplementary material, figure S4 shows a solution with $c_m = 0.2$ and GF = 40, where the 1:1 synchronization state can be seen along with phase portraits.

## 3.2. The effect of dexamethasone

We now simulate the effect of Dex-treatment. Dex induces PER expression and synchronizes clocks in populations of cells. This induction of PER is probably done via activation of the transcriptional activator glucocorticoid receptor (GR) [22]. Thus, to analyse its effect the constant term Dex is added to the equation of PER:CRY that in the reduced system includes the transcriptional terms of PER activation (see appendix A), as

$$\frac{d[\text{PER : CRY}]}{dt} = \text{Dex} + V_{D2}[\text{DBP}] - \gamma_{bp}[\text{BMAL1}][\text{PER : CRY}]. \tag{3.2}$$

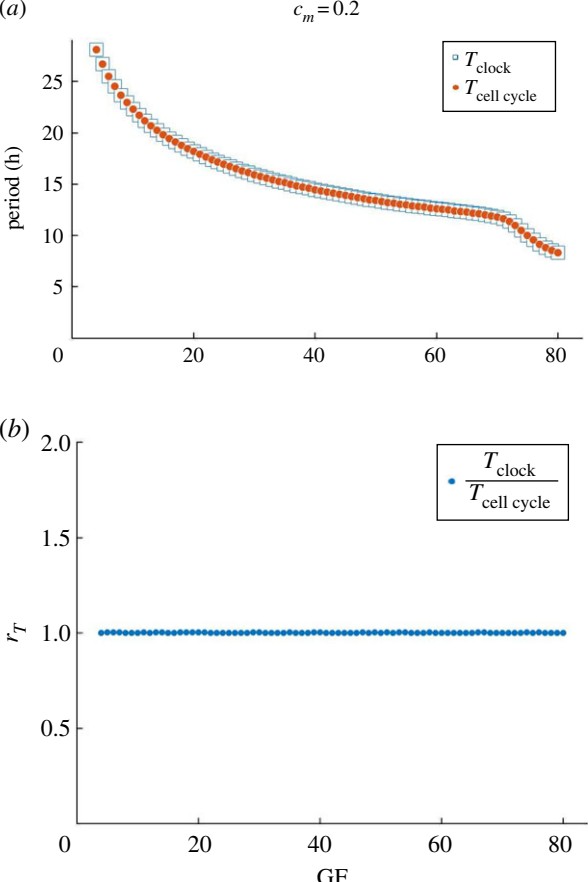

**Figure 2.** Strong coupling of the circadian clock and cell cycle models by MPF-induced degradation of REV. For $c_m = 0.2$, the system is strongly coupled with 1 : 1 period-lock. The period of the clock follows that of the cell cycle that decreases with GF. Cell cycle oscillations occur for $4 \leq \text{GF} \leq 80$.

With $c_m = 0.1$ the system couples in a 1 : 1 ratio for $4 \leq \text{GF} \leq 24$ and we verify that introducing Dex = 10 drives the system from the 1 : 1 to the 3 : 2 PL ratio, as shown in figure 5, confirming the ability of the model of reproducing the period-lock response to Dex.

These results mean that introducing an input on PER/PER : CRY has a similar effect to decreasing the coupling strength parameter $c_m$, in terms of driving the system from 1 : 1 PL to a higher PL ratio. The circadian clock dual state property (CLOCK : BMAL1 and PER : CRY in phase opposition) may be important to generate the wide range of period-lock ratios, as the introduction of Dex asymmetrically promotes one of the main clock phases (the state of high PER : CRY/low BMAL1). To further explore this phase opposition hypothesis, we devise a symmetric study where the system is in 2 : 1 PL and an input $I_B$ is applied to induce a high BMAL1 state. In contrast to the Dex input, we now anticipate the system will evolve to a state of 1 : 1 PL. The equation of BMAL1 becomes

$$\frac{\text{d[BMAL1]}}{\text{d}t} = I_B + V_R \frac{k_{Rr}^2}{k_{Rr}^2 + [\text{REV}]^2} - \gamma_{bp}[\text{BMAL1}][\text{PER} : \text{CRY}]. \tag{3.3}$$

In figure 6, we verify that our hypothesis is correct as the $I_B$ input drives the system from $r_T > 1$ to $r_T = 1$.

As coupling via BMAL1 is not being modelled, changes applied on the clock system impact the dynamics of the unidirectional **cell cycle → clock** coupled system because they affect the clock period. The electronic supplementary material, figure S5 shows that Dex increases the clock period, while $I_B$ decreases it. Thus, $I_B$ is promoting closer clock/cell cycle periods and consequently the 1 : 1 synchronization state, while Dex has an opposite effect. Promoting 1 : 1 PL either by increasing Dex (thus PER : CRY) or by decreasing $c_m$ (thus increasing REV), can lead to similar effects in BMAL1 as both PER : CRY and REV repress BMAL1. This result further suggests that the CLOCK : BMAL1 and PER : CRY molecular clock phases have opposite effects on the control of clock period and synchronization.

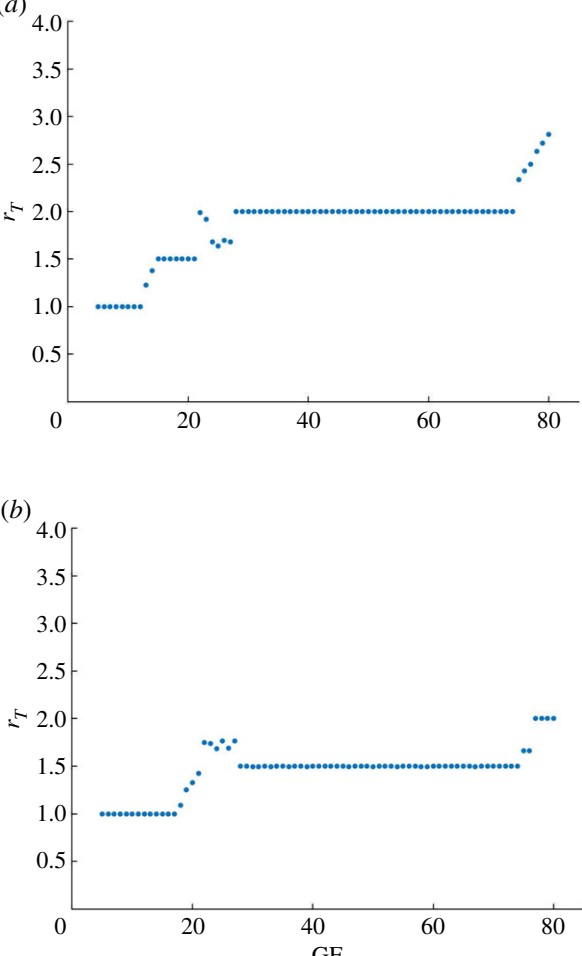

**Figure 3.** Weak coupling of the circadian clock and cell cycle models by MPF-induced degradation of REV. For (*a*) $c_m = 0.04$ and (*b*) $c_m = 0.08$, the system is in weak/moderate coupling and distinct period-lock ratios are obtained depending on GF, forming a pattern similar to that of the *devil's staircase*, where the period-lock ratio is increasing but remains constant by intervals of GF. GF and $c_m$ are control parameters for the PL ratios. The 3 : 2 experimentally observed PL state is obtained.

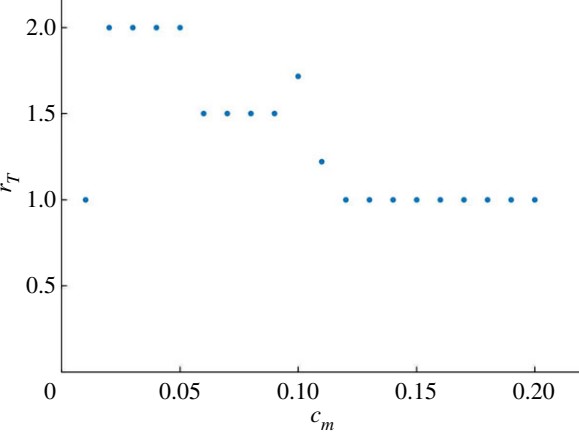

**Figure 4.** $c_m$ is a control parameter for the period-lock dynamics of the coupled system. Varying $c_m$ with fixed GF = 40 causes the ratio of clock to cell cycle period to vary in steps, where the 2 : 1, 3 : 2 and 1 : 1 period-lock ratios are obtained.

Furthermore, in this section, we have observed that for low GF values the oscillators tend to couple in a 1 : 1 fashion, when the clock and cell cycle intrinsic periods are closer. Accordingly, PL states with $r_T > 1$ that occur for higher GF values always represent a slower clock, which is in agreement with the experimental observations of Feillet *et al.* [11]. Moreover, the application of Dex is able to induce the

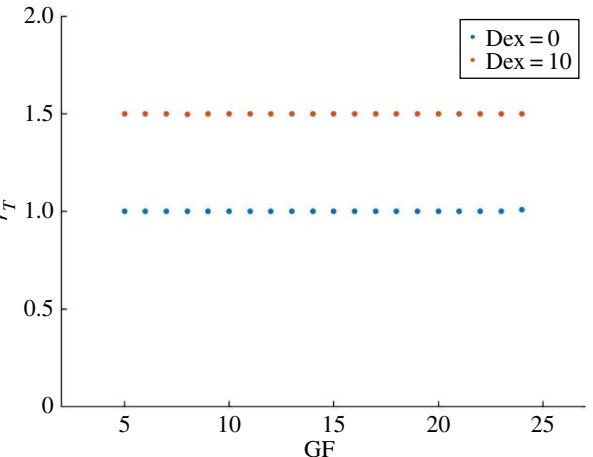

**Figure 5.** An input of Dex drives the system from $1:1$ to $3:2$ period-lock. With $c_m = 0.1$ and Dex $= 0$ the system is in strong coupling with $1:1$ PL for $4 \leq$ GF $\leq 24$. With $c_m = 0.1$ and Dex $= 10$ the $3:2$ PL ratio is obtained.

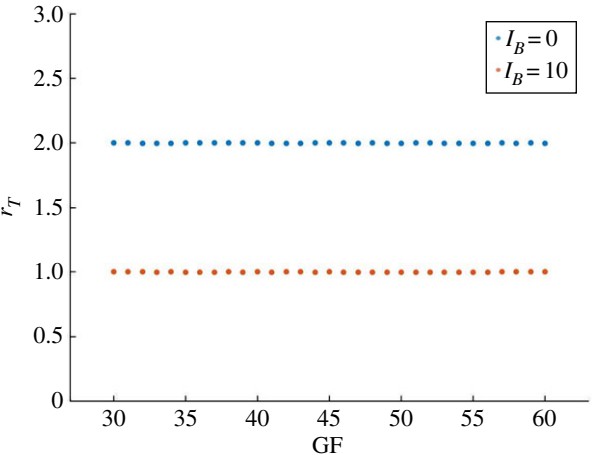

**Figure 6.** The input $I_B$ drives the system from $2:1$ to $1:1$ period-lock. With $c_m = 0.04$ and $I_B = 0$ the system period-locks in $2:1$ for $30 \leq$ GF $\leq 60$ (see also figure 3). With $c_m = 0.04$ and $I_B = 10$ the $1:1$ period-lock is obtained.

system from the $1:1$ to the $3:2$ PL ratio as in experimental observations by Feillet *et al.* [11]. These numerical experiments show that there is more than one way to drive the system between different PL states, suggesting that, in wild-type cells, the cell cycle may play an important role in regulating the clock period, as recently proposed by Feillet *et al.* [11].

# 4. Bidirectional coupling

The influence of the clock on the cell cycle has been known and documented for a long time. One notable mechanism and the one we will focus on here is the induction of the *wee1* gene by CLOCK : BMAL1 [5].

The action of WEE1 on MPF is included in the self-regulatory loop where MPF represses the negative loop representative of its inactivation by WEE1 (see §2) [18]. Thus, the term on $V_w$ is multiplied by a coupling term:

$$
\begin{aligned}
\frac{\mathrm{d}[MPF]}{\mathrm{d}t} = {} & \mathrm{GF} + V_c \frac{\overline{\mathrm{MPF_{max}}} - [\mathrm{MPF}]}{\mathrm{MPF_{max}} - [\mathrm{MPF}] + k_c} \frac{[\mathrm{MPF}]^2}{[\mathrm{MPF}]^2 + k_m^2} \\
& - c_b[\mathrm{BMAL1}]V_w \frac{[\mathrm{MPF}]}{[\mathrm{MPF}] + k_w} \frac{k_n^2}{[\mathrm{MPF}]^2 + k_n^2} \\
& - \gamma_1[\mathrm{APC} : \mathrm{cdc20}][\mathrm{MPF}],
\end{aligned}
\tag{4.1}
$$

where $c_b$ is the coupling strength parameter.

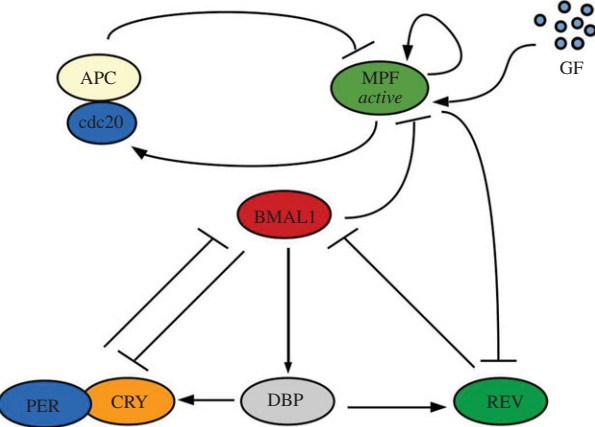

**Figure 7.** Schematic of the bidirectional coupling mechanism. Bidirectional coupling between the cell cycle and clock oscillators that includes two forms of coupling: MPF phosphorylates REV inducing its degradation and BMAL1 represses MPF by promoting *wee1*.

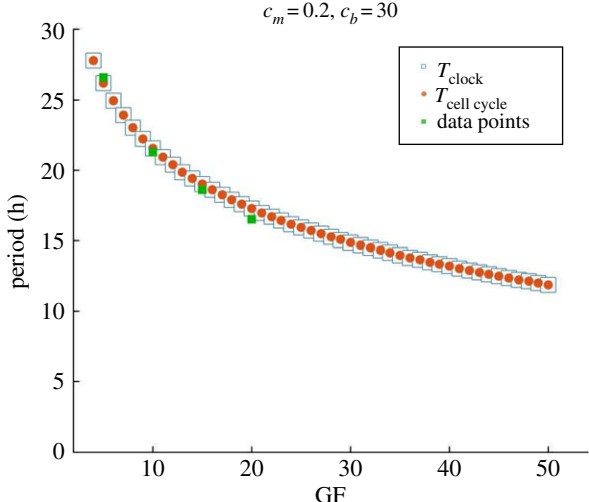

**Figure 8.** Period response of the bidirectional coupled system. Our bidirectional coupling mechanisms are able to reproduce the overall observed oscillators' period response of acceleration with GF and result in a good fit to experimental data [11,18], taking GF = %FBS, with $c_m = 0.2$ and $c_b = 30$.

Figure 7 shows a scheme of the system with bidirectional coupling.

## 4.1. The effect of growth factor

To verify that the system's period response to variations of GF is compatible with observations, figure 8 shows entrainment for values of coupling strength $c_m = 0.2$ and $c_b = 30$. The trend of period decrease with GF fits well to data points [11,18] when making a 1 to 1 correspondence GF = %FBS. We chose only the parameters $c_m$ and $c_b$ and maintained all remaining parameters calibrated as in table 1.

Furthermore, figure 9 shows the system's synchronization state for varying $c_m$ and $c_b$ with fixed GF = 20. Patterns of entrainment include the Arnold Tongue [23,24] for the 3 : 2 ratio. There is a blank region without oscillation, where the system converges to a steady state instead. Because GF = 20 causes a faster cell cycle than clock, synchronization states aside from the 1 : 1 tend to represent a slower clock than cell cycle.

## 4.2. The effect of dexamethasone

Next, we perform the Dex experiment, similarly to §3.2, by adding a constant amount of Dex in the system with 1 : 1 synchronization and observing its period-lock response. Figure 10 shows that the

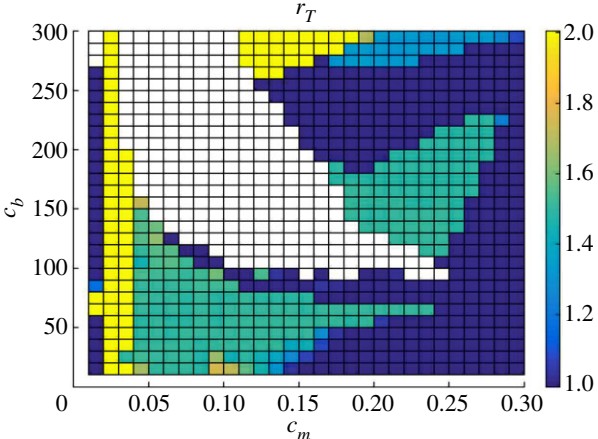

**Figure 9.** Period-lock for different values of $c_b$ and $c_m$ with GF = 20. Varying $c_b$ and $c_m$ for fixed GF = 20 results in different period-lock ratios. In the white region, there is no oscillation. The $1:1$, $3:2$, $2:1$ and $4:3$ ratios are the most prevalent. An Arnold Tongue pattern is visible for $3:2$ synchronization.

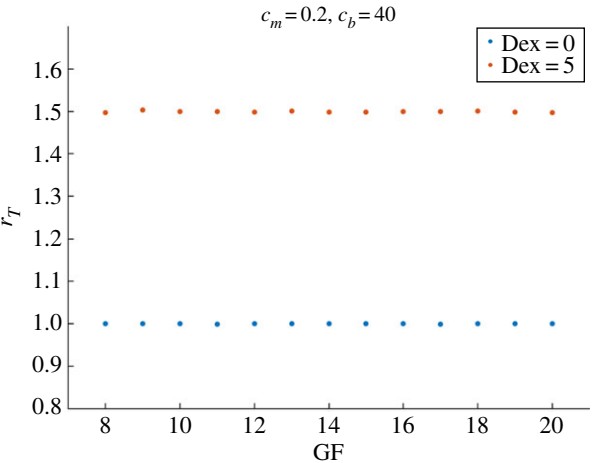

**Figure 10.** A Dex input induces the system from a $1:1$ to a $3:2$ period-lock in bidirectional coupling. With $c_m = 0.2$ and $c_b = 40$ the system locks in $1:1$ synchronization for $8 \leq$ GF $\leq 20$. Adding Dex = 5 shifts the system to a $3:2$ period-lock ratio for the same values of GF.

bidirectional coupling allows recovery of the experimentally observed effect of Dex [11] in shifting the system from $1:1$ to $3:2$ period-lock.

Dex is a control parameter for the oscillators' synchronization state. Additionally, the electronic supplementary material, figure S6 shows that Dex application has a nonlinear effect on the periods of the clock and cell cycle oscillators by contrast to its effect in increasing the clock period when in exclusive **cell cycle → clock** coupling (electronic supplementary material, figure S5).

## 4.3. Adding a dexamethasone pulse: implications for cell populations

So far, Dex has been introduced in the system as a constant input. However, to better reproduce experimental settings, where Dex is applied for some time (typically 30 min to 2 h) and then removed, we now apply a Dex pulse and observe the transient synchronization state change. We use the same parameters as in figure 10 ($c_m = 0.2$ and $c_b = 40$) with fixed GF = 15, which result in a $T_{\text{clock}} = T_{\text{cell cycle}} = 18.9$ h without Dex. A pulse of Dex = 40 is applied during 1 h and the synchronization response measured afterwards. This is, in a sense, a protocol similar to that of phase response curves on clocks, but instead, here we measure the period-lock ratios. Figure 11 shows the synchronization response to a pulse of Dex applied over the course of two periods and corresponds to two sequential response curves. $r_T$ is computed using the clock and cell cycle periods measured between the second and third cycles following the application of the pulse. The system's synchronization response is dependent on

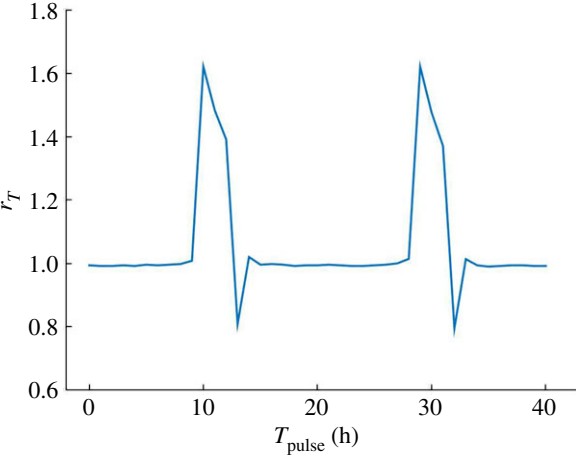

**Figure 11.** Response curves of the synchronization ratio of the bidirectional coupled system over two periods. A 1 h Dex pulse is applied over the course of two periods. Parameters are $c_m = 0.2$ and $c_b = 40$, $GF = 15$ and $Dex_{pulse} = 40$. The system's synchronization state changes only for certain times of pulse application $T_{pulse}$. The responsive phase corresponds to that of increasing BMAL1.

the time of pulse application $T_{pulse}$, i.e. the system responds by shifting away from the 1 : 1 period-lock only when the pulse is applied at particular clock/cell cycle phases. More specifically, we have verified that the Dex response peak occurs when BMAL1 is up. Furthermore, we verified that the strength and duration of Dex pulse both increase the maximum synchronization ratio achieved at the responsive phase.

As a population of cells contains cells that are asynchronous among themselves in their clock and cell cycle oscillations, this result provides insight on the observed existence of two groups of cells after Dex application mentioned in §1. Namely, that the existence of two period-lock groups of cells observed by Feillet *et al.* [11]—one with cells locking in a 1 : 1 manner and the other with cells locking in a 3 : 2 manner—may be owing to the clock or cell cycle phase cells are in at the moment of Dex application. Moreover, a small region of period-lock with $r_T < 1$ occurs in figure 11, predicting that, for a small phase interval, a Dex application could result in a slower cell cycle than clock, thus providing a strategy of cell cycle period control.

It is not clear, in theory, whether the clock or the cell cycle would be the preponderant oscillator in determining the synchronization state response. On the one hand, cells in the G0 phase of the cell cycle are known to be unresponsive to a variety of inputs and cells in M phase do not respond to Dex-treatment [25]. On the other hand, not only Dex is known to induce *Per* [22] but also a recent crosstalk between REV-ERB$\alpha$ and GR signalling has been uncovered [26]. Therefore, both systems could play a role in the resulting synchronization state of a cell under the application of Dex or other inputs. In our work, we found that the system is Dex-responsive at the molecular phase where BMAL1 is up and PER : CRY is down. This makes sense intuitively, as it is when PER : CRY is down that an additive input in its expression would result in bigger changes, whereas when PER : CRY is already up adding more won't affect the system as much. From this analysis, the time of pulse application $T_{pulse}$ is a control parameter for the oscillators' synchronization state.

Lastly, the electronic supplementary material, figure S7 shows the same simulation of figure 11 at a later point in time (1000 h after the pulse), showing that after a transient shift in synchronization state, the system returns to the 1 : 1 period-lock. As mentioned in §1, the different period-lock ratios observed by Feillet *et al.* were considered to be caused by the existence of multiple attractors [11], i.e. that the change caused by Dex occurs because the system shifts to a different limit-cycle, with each ratio of period-lock corresponding to a different attractor. In this regard, our results add the hypothesis of the Dex-induced period-lock change being transient. Experimental observation of cells after the application of the Dex pulse was done over the course of 3 days [11], which can easily fall within the transient period. For this reason, a longer observational time would help to clarify the existence (or non-existence) of attractors.

Finally, our observations also indicate that a lasting effect of altered synchronization state can be achieved by means of a constant Dex input (of a smaller value than that given in a pulse).

# 5. Final discussion

In this work, we have modelled the coupling between clock and cell cycle in mammalian cells and analysed the oscillators' synchronization dynamics. We have observed entrainment, the emergence of rational clock to cell cycle period ratios and how control parameters drive the system between different synchronization states. In particular, variations of control parameters often result in a *devil's staircase* pattern for the synchronization ratios. Additionally, the effect of GF on the periods of cell cycle and clock in 1 : 1 entrainment adequately compares to data points.

Besides increasing the understanding of the coupled mammalian clock/cell cycle system, this analysis facilitates the discovery of novel system control tools. For instance, an immediate physiological benefit is to be able to control the period of one oscillator using inputs that affect the other. An example of this is provided by our previous work (Almeida *et al.* [20]) where slowing down the cell cycle (which has implications for cancer control) was successfully achieved by tuning clock parameters. Moreover, there are already established compounds with a modulating effect on some of these parameters, such as GSk3$\beta$-inhibiting drugs, known to increase phosphorylation of REV-ERB, that can lead either to a decreased or increased clock period [27] depending on the GSK-inhibitor used [28,29].

An important observation is that GF is a control parameter to the oscillators' synchronization state. This, in theory, predicts that synchronization states differing from 1 : 1, such as the experimental results of Feillet *et al.* [11], could be reproduced by further increasing GF. However, this verification is contingent upon the GF region of $r_T \neq 1$ falling within the physiological limit of GF increase in cells. Nevertheless, it would be interesting to analyse synchronization in cells grown in high GF (%FBS > 15) without Dex, as experimental observations of the 20% FBS culture were made only in the presence of Dex [11]. On the other hand, if the natural coupling between the oscillators is strong enough the oscillators remain in 1 : 1 period-lock even for increasing GF.

Observations of a Dex pulse on the bidirectional coupled system reveal a sensitivity of the CLOCK : BMAL1 molecular phase in synchronization state response by comparison to the PER : CRY phase (that is irresponsive to Dex), which we relate to the existence of two populations of cells observed by Feillet *et al.* [11]. This insight might further be relevant for the understanding of a variety of chemical therapies, including apoptosis-inducing chemotherapies that are usually efficient for only a part of the cell population, with a percentage of cells remaining resistant. In this regard, though a period response differs from an apoptotic response our results uncover a clock and cell cycle time-dependent response to inputs, which may help to explain the increased efficiency demonstrated by chronotherapies over normal therapies. Thus, we have found that not only the amount and duration of the Dex pulse are control parameters for the system's dynamical response, but also the time of pulse application as it relates to the clock/cell cycle oscillators' phase.

Moreover, we found the system's response to a Dex pulse to be transient. A similar conclusion to ours is given by Traynard *et al.* [14], albeit in the context of non-recovery of experimental period-lock ratios. By contrast, in this work, we verify the occurrence of specific experimental synchronization ratios in agreement with the work of Feillet *et al.* [11], while at the same time demonstrating that these ratios might be transient when occurring in response to a pulse. Thus, here we consider the occurrence of rational synchronization ratios and the existence of multiple attractors as two different questions.

Furthermore, we have verified the synchronization state to be dependent on the relationship between the intrinsic periods of the two oscillators, as well as on the coupling strength. Specifically, 1 : 1 synchronization can be obtained when the intrinsic periods between the oscillators are close or when the coupling strength is high. In particular, inputs such as GF and Dex induce changes in synchronization state by changing the proximity between the periods of the oscillators. Dex induces a change similar to that of GF, because these two inputs both separate the period of the oscillators: GF by accelerating the cell cycle and Dex by slowing down the clock. As a result, we have observed *in silico* that in the presence of Dex there is a decrease of the GF value required for a change of synchronization ratio, which provides an explanation to the experimental observations of Feillet *et al.* [11].

Dex application as a PER/PER : CRY input recovered the experimentally observed change in synchronization state ($\{r_T = 1\} \rightarrow \{r_T > 1\}$) [11] and an input $I_B$ applied on BMAL1 had the opposite effect of Dex ($\{r_T > 1\} \rightarrow \{r_T = 1\}$). This is because inputs at the two main clock phases induce opposite effects on clock period. Thus, the circadian clock topology recovering CLOCK : BMAL1/PER : CRY antiphase is relevant for the change in synchronization state induced by Dex application or by the application of inputs that asymmetrically promote one of the two main clock phases.

Lastly, our main idea concerning unidirectional cell cycle → clock coupling is that MPF phosphorylates an essential clock component. The particular mechanism modelled here was that of MPF-induced REV degradation (observed experimentally [17]), which resulted in entrainment and allowed to recover the effect of Dex application. These results suggest the viability of a class of coupling mechanisms involving the phosphorylation of a core clock component by the essential cell cycle machinery.

Data accessibility. This article has no additional data.
Authors' contributions. S.A. carried out the modelling and simulation work, participated in the conception and design of the study and wrote the manuscript. M.C. coordinated the study, participated in its conception and design and revised the manuscript. F.D. participated in the conception and design of the study and revised the manuscript. All authors gave final approval for publication and agree to be held accountable for the work performed therein.
Competing interests. We declare we have no competing interests.
Funding. The authors are part of the Labex SIGNALIFE Network for Innovation on signal Transduction Pathways in Life Sciences (grant no. ANR-11-LABX-0028-01) and ICycle project (ANR-16-CE33-0016-01).

# Appendix A. Reduction of the mammalian clock model

The mammalian cellular clock model of Almeida *et al.* [19] is given by equations:

$$\frac{d[BMAL1]}{dt} = R_{box} - \gamma_{bp}[BMAL1][PER:CRY] \tag{A 1}$$

$$\frac{d[ROR]}{dt} = E_{box} + R_{box} - \gamma_{ror}[ROR] \tag{A 2}$$

$$\frac{d[REV]}{dt} = 2E_{box} + D_{box} - \gamma_{rev}[REV] \tag{A 3}$$

$$\frac{d[DBP]}{dt} = E_{box} - \gamma_{db}[DBP] \tag{A 4}$$

$$\frac{d[E4BP4]}{dt} = 2R_{box} - \gamma_{E4}[E4BP4] \tag{A 5}$$

$$\frac{d[CRY]}{dt} = E_{box} + 2R_{box} - \gamma_{pc}[PER][CRY] + \gamma_{cp}[PER:CRY] - \gamma_{c}[CRY] \tag{A 6}$$

$$\frac{d[PER]}{dt} = E_{box} + D_{box} - \gamma_{pc}[PER][CRY] + \gamma_{cp}[PER:CRY] - \gamma_{p}[PER] \tag{A 7}$$

and

$$\frac{d[PER:CRY]}{dt} = \gamma_{pc}[PER][CRY] - \gamma_{cp}[PER:CRY] - \gamma_{bp}[BMAL1][PER:CRY] \tag{A 8}$$

where

$$E_{box} = V_E \frac{[BMAL1]}{[BMAL1] + k_E + k_{Er}[BMAL1][CRY]} \tag{A 9}$$

$$R_{box} = V_R \frac{[ROR]}{[ROR] + k_R} \frac{k_{Rr}^2}{k_{Rr}^2 + [REV]^2} \tag{A 10}$$

and

$$D_{box} = V_D \frac{[DBP]}{[DBP] + k_D} \frac{k_{Dr}}{k_{Dr} + [E4BP4]}. \tag{A 11}$$

In order to obtain the core structural dynamical network of this system, we perform a sequence of quasi-steady-state approximations and other simplifications, verifying at each step both the existence of a periodic solution and antiphase between PER:CRY and BMAL1. The goal is to reduce the number of variables and possibly simplify equation terms in order to obtain a skeleton model with dynamical properties similar to those of the original system [19].

We start by setting E4BP4 at the quasi-steady-state

$$\frac{d[E4BP4]}{dt} = 0, \tag{A 12}$$

which leads to a system without loss of oscillations and allows us to approximate the Michaelis–Menten term with a negative effect on $D_{box}$ ($k_{Dr}/(k_{Dr} + [E4BP4])$) (see equation (A 9)) by a constant. E4BP4 is

**Table 2.** Parameters of the reduced clock model.

| $p$ | numerical value |
| --- | --- |
| $V_R$ | 44.4%. h$^{-1}$ |
| $k_{Rr}$ | 80.1% |
| $V_B$ | 0.142%. h$^{-1}$ |
| $V_{D2}$ | 19.0%. h$^{-1}$ |
| $\gamma_{rev}$ | 0.241 h$^{-1}$ |
| $\gamma_{db}$ | 0.156 h$^{-1}$ |
| $\gamma_{bp}$ | 2.58%$^{-1}$. h |

smaller but close to $k_{Dr}$ [19], varying up to 86, with $k_{Dr}$ = 94.7, for which an approximative value of this $D_{box}$ term is (1/2). Furthermore, as $k_D \gg$ [DBP], the Michaelis–Menten term with a positive effect on $D_{box}$ ($V_D$ ([DBP])/([DBP] + $k_D$)) can be approximated by a linear function, which leads to the equation of $D_{box}$ being well approximated by

$$D_{box} = \frac{1}{2} \frac{V_D}{k_D} [DBP]. \tag{A 13}$$

Secondly, setting the equation for the formation of ROR at the quasi-steady-state

$$\frac{d[ROR]}{dt} = 0, \tag{A 14}$$

also maintains the desired oscillatory properties and eliminates one more variable, with the ROR term being approximated by 1. REV, however, cannot be removed, and as such $R_{box}$ can now be simplified as:

$$R_{box} = V_R \frac{k_{Rr}^2}{k_{Rr}^2 + [REV]^2}. \tag{A 15}$$

As a third step, the system does not require oscillation of CRY and we also verify the dependence of $E_{box}$ on CRY can be set to zero ($k_{Er}$ [BMAL1][CRY] = 0 in equation (A 7)). Furthermore, $k_E \gg$ [BMAL1] allows the following approximation for $E_{box}$:

$$E_{box} = \frac{V_E}{k_E} [BMAL1]. \tag{A 16}$$

Finally, consider the quasi-steady-state approximation of equation (A 7):

$$\frac{d[PER]}{dt} = 0, \tag{A 17}$$

that leads to

$$PER = \frac{E_{box} + D_{box} + \gamma_{cp}[PER:CRY]}{\gamma_{pc}[CRY] + \gamma_p}, \tag{A 18}$$

furthermore it is also possible to take

$$\gamma_p = 0, \tag{A 19}$$

which impacts the period of the system, but not the existence of oscillations. From (A 18) and (A 19), replacing PER in the PER:CRY equation (A 8) leads to cancelling out the term $\gamma_{pc}$[CRY] and dependence on CRY is automatically eliminated.

The reduced model has now four variables: BMAL1, DBP, REV and PER:CRY. We further observe that the $E_{boxes}$ in the equations of REV and of PER:CRY can be removed, while preserving oscillation and antiphasic relation between BMAL1 and PER:CRY, but the $D_{boxes}$ cannot. Finally, the skeleton reduced model is given by equations (2.3) to (2.6) in §2, where $V_B = (V_E/k_E)$ and $V_{D2} = (1/2)(V_D/k_D)$ and all parameters (now shown in table 2) come directly from the parameters of table 1 of Almeida

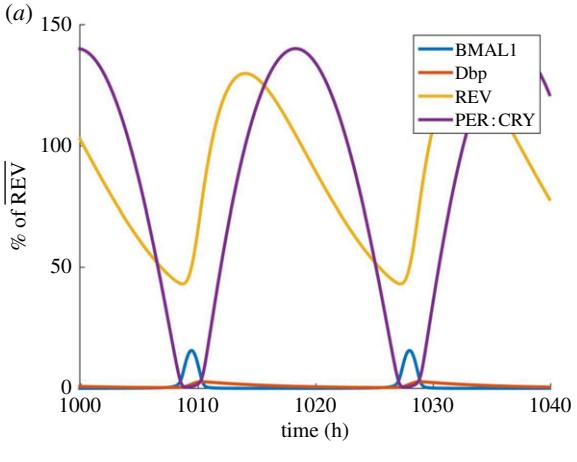

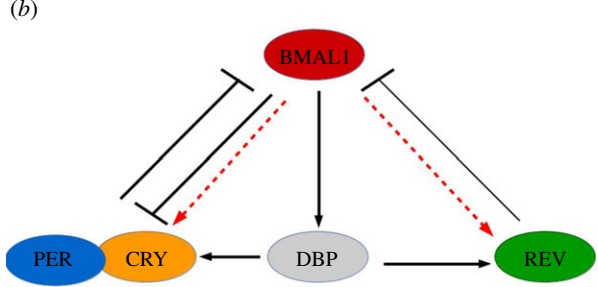

**Figure 12.** The reduced model can recover the main properties of the circadian clock. (*a*) Output of the reduced model, oscillations have a period of 18.6 h with parameters of table 2; BMAL1 and PER : CRY maintain an antiphasic oscillation. (*b*) A scheme of the reduced model. Red dashed arrows show the effect of $E_{\text{boxes}}$ on REV and PER : CRY and can be removed.

*et al.* [19]. The boxes here have become $R_{\text{box}} = V_R(k_{Rr}^2)/(k_{Rr}^2 + [\text{REV}]^2)$, $E_{\text{box}} = V_B [\text{BMAL1}]$ and $D_{\text{box}} = V_{D2}$ [DBP].

Figure 12*a* shows a simulation of the reduced model: the solution has an oscillatory period of 18.6 h and BMAL1 and PER : CRY maintaining an antiphasic relation. Figure 12*b* shows a scheme of the reduced model, the dashed red line represents the $E_{\text{boxes}}$ at PER and REV promoter that can be removed as in equations (2.5) and (2.6); the direct double negative loop between BMAL1 and PER : CRY is owing to their mutual removal from gene promoters as the [CLOCK : BMAL1] : [PER : CRY] complex.

Our final scheme suggests that in a reduced skeleton model approach, maintaining antiphasic BMAL1/PER : CRY oscillation requires a network topology of two alternative pathways for the action of BMAL1 on the PER : CRY complex, allowing for the possibility of delay effects and distinct time-scales phenomena. In this model, this role is achieved by $D_{\text{box}}$.

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
