## [Reviewer comments · Royal Society Open Science]

Review History

RSOS-192054.R0 (Original submission)

Review form: Reviewer 1 (Annabelle Ballesta)

Is the manuscript scientifically sound in its present form?

Yes

Are the interpretations and conclusions justified by the results?

Yes

Is the language acceptable?

Yes

Do you have any ethical concerns with this paper?

No

Have you any concerns about statistical analyses in this paper?

No

Recommendation?

Accept with minor revision (please list in comments)

Comments to the Author(s)

Almeida et al. performed a modeling study to investigate the molecular links between the circadian clock and the cell cycle at the level of a single cell. This work builds on simplified ODE-based models for the clock and the cell cycle that they previously developed. Here, they present a focus on the influence of the cell cycle on the circadian clock and on the bidirectional interactions between the two oscillators. The developed coupled model successfully recapitulates biological information about phase locking and period changes in response to various stimuli. Moreover, the authors have placed their work in regards to the existing modeling works. The article is clear and well written. Here are minor concerns:

-How were initial conditions of protein chosen? What is their Impact on the main results?

-How were the model parameters (Table I) chosen? Have the authors used qualitative or quantitative information? In particular, for Figure 8, how were estimated parameters to obtain such a good fit to data? Which parameters were chosen to fit data? C_m/ c_b ? kinetics parameters?

-It seems that the proposed mathematical formulation for model coupling implies that each oscillator does not oscillate on its own when coupling parameters are set to zeros. For instance, in equation 7, no degradation of $Reverb\text{-}\alpha$ occurs if MPF is set to zero which probably impairs oscillations. This is not in agreement with biological literature as non-cycling, i.e. quiescent cells, do have a functional clock. Please better justify this choice of modeling and discuss its advantages/disadvantages

-page 8 line 44 "Dex can either decrease or increase the period of both clock and cell cycle oscillators in the system with bidirectional coupling, in contrast to its effect in increasing the clock period when in exclusive cell cycle/clock coupling (Supporting Fig. 5)." Please reformulate this sentence which is misleading as Dex exposure results in a period greater than 24h for all simulated concentrations. However, the relationship between the dose and the periods' length is indeed not linear.

-Figure 2 and Figure 8, please increase size of pictograms in the legends, as they it is difficult to distinguish between squares and bullets as it is.

-Page 8, line 3: include a reference for Arnold tongue patterns

Review form: Reviewer 2

Is the manuscript scientifically sound in its present form?

Yes

Are the interpretations and conclusions justified by the results?

Yes

Is the language acceptable?

Yes

Do you have any ethical concerns with this paper?

No

Have you any concerns about statistical analyses in this paper?

No

Recommendation?

Accept with minor revision (please list in comments)

Comments to the Author(s)

The manuscript "Control of synchronization ratios in clock/cell cycle coupling by growth factor and glucocorticoids" uses a mathematical model to explore the dynamical mechanism of multiple phase-locked behaviors for cell cycle and circadian clock. The authors reproduce some experimental observations in the recent relevant studies (Feillet et al). Moreover, they study a bidirectional coupling system of cell cycle and circadian clock, and analyze the synchronization ratios upon Dex pulse. This manuscript addresses a very interesting topic in coupled oscillatory systems. I would recommend the manuscript for publishing in Royal Society Open Science after the issues raised below are resolved.

1. One problem of the manuscript is that there is little discussion of the physiological significance of the existence of multiple attractors for cell cycle and cell cycle phase-locked behavior. Are there any physiological benefits if the synchronization ratios can controlled by growth factor and glucocorticoids.
2. What happens in the blank region of Fig.9? And why the period-lock ratio is always larger than 1?
3. The manuscript claims that the system can shift away from the 1:1 period-lock behavior only when the Dex pulse is applied at particular clock/cell cycle phases. It would be clearer that the authors show a related phase response curve. Also, how the duration and the strength of the DEX pulse affect the phase response curve?

Decision letter (RSOS-192054.R0)

02-Jan-2020

Dear Dr Almeida

On behalf of the Editors, I am pleased to inform you that your Manuscript RSOS-192054 entitled "Control of Synchronization Ratios in Clock/Cell Cycle Coupling by Growth Factors and Glucocorticoids" has been accepted for publication in Royal Society Open Science subject to minor revision in accordance with the referee suggestions. Please find the referees' comments at the end of this email.

The reviewers and handling editors have recommended publication, but also suggest some minor revisions to your manuscript. Therefore, I invite you to respond to the comments and revise your manuscript.

- Ethics statement

- Data accessibility

It is a condition of publication that all supporting data are made available either as supplementary information or preferably in a suitable permanent repository. The data accessibility section should state where the article's supporting data can be accessed. This section should also include details, where possible of where to access other relevant research materials

such as statistical tools, protocols, software etc can be accessed. If the data has been deposited in an external repository this section should list the database, accession number and link to the DOI for all data from the article that has been made publicly available. Data sets that have been deposited in an external repository and have a DOI should also be appropriately cited in the manuscript and included in the reference list.

If you wish to submit your supporting data or code to Dryad (<http://datadryad.org/>), or modify your current submission to dryad, please use the following link:
<http://datadryad.org/submit?journalID=RSOS&manu=RSOS-192054>

- **Competing interests**

- **Authors' contributions**

- **Acknowledgements**

- **Funding statement**

Because the schedule for publication is very tight, it is a condition of publication that you submit the revised version of your manuscript before 11-Jan-2020. Please note that the revision deadline will expire at 00.00am on this date. If you do not think you will be able to meet this date please let me know immediately.

When submitting your revised manuscript, you will be able to respond to the comments made by

the referees and upload a file "Response to Referees" in "Section 6 - File Upload". You can use this to document any changes you make to the original manuscript. In order to expedite the processing of the revised manuscript, please be as specific as possible in your response to the referees. We strongly recommend uploading two versions of your revised manuscript:

If your manuscript is newly submitted and subsequently accepted for publication, you will be asked to pay the article processing charge, unless you request a waiver and this is approved by Royal Society Publishing. You can find out more about the charges at <https://royalsocietypublishing.org/rsos/charges>. Should you have any queries, please contact openscience@royalsociety.org.

on behalf of Professor Mark Chaplain (Subject Editor)
 openscience@royalsociety.org

Associate Editor Comments to Author (Professor Mark Chaplain):

Please address the points raised by both reviewers in a revised version. Thank you.

Reviewer comments to Author:

Reviewer: 1

Comments to the Author(s)

Almeida et al. performed a modeling study to investigate the molecular links between the circadian clock and the cell cycle at the level of a single cell. This work builds on simplified ODE-based models for the clock and the cell cycle that they previously developed. Here, they present a focus on the influence of the cell cycle on the circadian clock and on the bidirectional interactions between the two oscillators. The developed coupled model successfully recapitulates biological information about phase locking and period changes in response to various stimuli. Moreover, the authors have placed their work in regards to the existing modeling works. The article is clear and well written. Here are minor concerns:

-How were initial conditions of protein chosen? What is their Impact on the main results?

-How were the model parameters (Table I) chosen? Have the authors used qualitative or quantitative information? In particular, for Figure 8, how were estimated parameters to obtain such a good fit to data? Which parameters were chosen to fit data? C_m / c_b ? kinetics parameters?

-It seems that the proposed mathematical formulation for model coupling implies that each oscillator does not oscillate on its own when coupling parameters are set to zeros. For instance, in equation 7, no degradation of $Reverb$ - α occurs if MPF is set to zero which probably impairs oscillations. This is not in agreement with biological literature as non-cycling, i.e. quiescent cells, do have a functional clock. Please better justify this choice of modeling and discuss its advantages/disadvantages

-page 8 line 44 "Dex can either decrease or increase the period of both clock and cell cycle oscillators in the system with bidirectional coupling, in contrast to its effect in increasing the clock period when in exclusive cell cycle/clock coupling (Supporting Fig. 5)." Please reformulate this sentence which is misleading as Dex exposure results in a period greater than 24h for all simulated concentrations. However, the relationship between the dose and the periods' length is indeed not linear.

-Figure 2 and Figure 8, please increase size of pictograms in the legends, as they it is difficult to distinguish between squares and bullets as it is.

-Page 8, line 3: include a reference for Arnold tongue patterns

Reviewer: 2

Comments to the Author(s)

The manuscript "Control of synchronization ratios in clock/cell cycle coupling by growth factor and glucocorticoids" uses a mathematical model to explore the dynamical mechanism of multiple phase-locked behaviors for cell cycle and circadian clock. The authors reproduce some experimental observations in the recent relevant studies (Feillet et al). Moreover, they study a

bidirectional coupling system of cell cycle and circadian clock, and analyze the synchronization ratios upon Dex pulse. This manuscript addresses a very interesting topic in coupled oscillatory systems. I would recommend the manuscript for publishing in Royal Society Open Science after the issues raised below are resolved.

1. One problem of the manuscript is that there is little discussion of the physiological significance of the existence of multiple attractors for cell cycle and cell cycle phase-locked behavior. Are there any physiological benefits if the synchronization ratios can be controlled by growth factor and glucocorticoids.
2. What happens in the blank region of Fig.9? And why the period-lock ratio is always larger than 1?
3. The manuscript claims that the system can shift away from the 1:1 period-lock behavior only when the Dex pulse is applied at particular clock/cell cycle phases. It would be clearer that the authors show a related phase response curve. Also, how the duration and the strength of the DEX pulse affect the phase response curve?

Author's Response to Decision Letter for (RSOS-192054.R0)

See Appendix A.

Decision letter (RSOS-192054.R1)

17-Jan-2020

Dear Dr Almeida,

It is a pleasure to accept your manuscript entitled "Control of Synchronization Ratios in Clock/Cell Cycle Coupling by Growth Factors and Glucocorticoids" in its current form for publication in Royal Society Open Science. The comments of the reviewer(s) who reviewed your manuscript are included at the foot of this letter.

on behalf of Professor Mark Chaplain (Associate Editor) and Mark Chaplain (Subject Editor)
openscience@royalsociety.org

Appendix A

Response to Reviews

Reviewer: 1

Almeida et al. performed a modeling study to investigate the molecular links between the circadian clock and the cell cycle at the level of a single cell. This work builds on simplified ODE-based models for the clock and the cell cycle that they previously developed. Here, they present a focus on the influence of the cell cycle on the circadian clock and on the bidirectional interactions between the two oscillators. The developed coupled model successfully recapitulates biological information about phase locking and period changes in response to various stimuli. Moreover, the authors have placed their work in regards to the existing modeling works. The article is clear and well written. Here are minor concerns:

-How were initial conditions of protein chosen? What is their impact on the main results?

For both oscillators any initial conditions, including making all variables zero, converge to the oscillatory solution. They don't remain at zero because there are additive terms in each oscillator: GF in the MPF variable of the cell cycle and the R-box term that becomes VR if REV is zero for the BMAL1 equation in the cell cycle. Thus, initial conditions used in simulations were tentatively chosen. We added this information on page 3.

-How were the model parameters (Table I) chosen? Have the authors used qualitative or quantitative information? In particular, for Figure 8, how were estimated parameters to obtain such a good fit to data? Which parameters were chosen to fit data? C_m / c_b ? kinetics parameters?

Parameters were obtained by non-linear optimization based on cost-minimization, with the use of a MATLAB function. This calibration was performed in the development of the clock and cell cycle models. We added this information on page 3. For figure 8, we chose only the c_m and c_b parameters and maintained all remaining parameters calibrated as in Table I, we reinforce this on page 6.

-It seems that the proposed mathematical formulation for model coupling implies that each oscillator does not oscillate on its own when coupling parameters are set to zeros. For instance, in equation 7, no degradation of Reverb- α occurs if MPF is set to zero which probably impairs oscillations. This is not in agreement with biological literature as non-cycling, i.e. quiescent cells, do have a functional clock. Please better justify this choice of modeling and discuss its advantages/disadvantages.

When an oscillator is knocked-down it isn't necessarily the case that all essential proteins go to zero, but rather to a constant value. Thus, it is possible to knock-down the cell cycle, as in quiescent cells, but not having all clock proteins at zero. Nevertheless if we assume that in non-dividing cells MPF is probably zero, we may say that it is a consequence of our reduced modeling approach to have a trade-off between having a reduced number of variables and being able to simulate de-coupling.

In a more complex model, knocking down MPF would affect the formation of an intermediate species [MPF:REV] and not act on the REV equation directly. The advantage of our type of modeling is to keep the minimum number of variables necessary to answer our scientific questions, concerning period synchronization.

-page 8 line 44 “Dex can either decrease or increase the period of both clock and cell cycle oscillators in the system with bidirectional coupling, in contrast to its effect in increasing the clock period when in exclusive cell cycle/clock coupling (Supporting Fig. 5).” Please reformulate this sentence which is misleading as Dex exposure results in a period greater than 24h for all simulated concentrations. However, the relationship between the dose and the periods’ length is indeed not linear.

We have reformulated this sentence as: “Dex application has a non-linear effect on the periods of the clock and cell cycle oscillators.” on page 6.

-Figure 2 and Figure 8, please increase size of pictograms in the legends, as they it is difficult to distinguish between squares and bullets as it is.

Ok. We have increased the squares and bullets on the legend box of these 2 Figures.

-Page 8, line 3: include a reference for Arnold tongue patterns

We added references 23 and 24, now cited on page 7.

Reviewer: 2

The manuscript “Control of synchronization ratios in clock/cell cycle coupling by growth factor and glucocorticoids” uses a mathematical model to explore the dynamical mechanism of multiple phase-locked behaviors for cell cycle and circadian clock. The authors reproduce some experimental observations in the recent relevant studies (Feillet et al). Moreover, they study a bidirectional coupling system of cell cycle and circadian clock, and analyze the synchronization ratios upon Dex pulse. This manuscript addresses a very interesting topic in coupled oscillatory systems. I would recommend the manuscript for publishing in Royal Society Open Science after the issues raised below are resolved.

1. One problem of the manuscript is that there is little discussion of the physiological significance of the existence of multiple attractors for cell cycle and cell cycle phase-locked behavior. Are there any physiological benefits if the synchronization ratios can controlled by growth factor and glucocorticoids.

Physiological benefits include being able to control the period one oscillator using inputs that affect the other. An example of this is our work Almeida et al., (2019), (reference 20) where slowing down the cell cycle (which may have implications for cancer control) was successfully achieved by tuning clock parameters. Moreover, there are already established compounds with a modulating effect on some of these parameters. We added a commentary discussing this topic on page 8 in the final discussion.

2. What happens in the blank region of Fig.9? And why the period-lock ratio is always larger than 1?

In the blank region there is no oscillation, but a steady-state instead, i.e. all protein values converge to a constant. We added this information on page 6.

The intrinsic periods of clock and cell cycle for $GF = 20$ are such that the clock's period is larger than the cell cycle period, therefore there is a tendency for the period-lock ratio to be larger than 1.

3. The manuscript claims that the system can shift away from the 1:1 period-lock behavior only when the Dex pulse is applied at particular clock/cell cycle phases. It would be clearer that the authors show a related phase response curve. Also, how the duration and the strength of the DEX pulse affect the phase response curve?

Fig. is a phase response curve over the course of 2 periods. We added this information in the figure legend and on page 7. The strength and duration of Dex pulse both increase the maximum synchronization ratio achieved at the responsive phase. We added this information on page 7.